# Fibromyalgia and Irritable Bowel Syndrome Interaction: A Possible Role for Gut Microbiota and Gut-Brain Axis

**DOI:** 10.3390/biomedicines11061701

**Published:** 2023-06-13

**Authors:** Cinzia Garofalo, Costanza Maria Cristiani, Sara Ilari, Lucia Carmela Passacatini, Valentina Malafoglia, Giuseppe Viglietto, Jessica Maiuolo, Francesca Oppedisano, Ernesto Palma, Carlo Tomino, William Raffaeli, Vincenzo Mollace, Carolina Muscoli

**Affiliations:** 1Department of Experimental and Clinical Medicine, “Magna Græcia” University of Catanzaro, 88100 Catanzaro, Italy; 2Department of Medical and Surgical Sciences, Neuroscience Research Center, “Magna Græcia” University of Catanzaro, 88100 Catanzaro, Italy; 3Physiology and Pharmacology of Pain, IRCCS San Raffaele Roma, 00166 Rome, Italy; 4Department of Health Science, Institute of Research for Food Safety & Health (IRC-FSH), “Magna Græcia” University of Catanzaro, 88100 Catanzaro, Italy; 5Scientific Direction, IRCCS San Raffaele Roma, 00166 Rome, Italy; 6Institute for Research on Pain, ISAL Foundation, Torre Pedrera, 47922 Rimini, Italy

**Keywords:** fibromyalgia, IBS, microbiota gut-brain axis

## Abstract

Fibromyalgia (FM) is a serious chronic pain syndrome, characterised by muscle and joint stiffness, insomnia, fatigue, mood disorders, cognitive dysfunction, anxiety, depression and intestinal irritability. Irritable Bowel Syndrome (IBS) shares many of these symptoms, and FM and IBS frequently co-exist, which suggests a common aetiology for the two diseases. The exact physiopathological mechanisms underlying both FM and IBS onset are unknown. Researchers have investigated many possible causes, including alterations in gut microbiota, which contain billions of microorganisms in the human digestive tract. The gut-brain axis has been proven to be the link between the gut microbiota and the central nervous system, which can then control the gut microbiota composition. In this review, we will discuss the similarities between FM and IBS. Particularly, we will focus our attention on symptomatology overlap between FM and IBS as well as the similarities in microbiota composition between FM and IBS patients. We will also briefly discuss the potential therapeutic approaches based on microbiota manipulations that are successfully used in IBS and could be employed also in FM patients to relieve pain, ameliorate the rehabilitation outcome, psychological distress and intestinal symptoms.

## 1. Introduction

The fibromyalgia syndrome (FM) is the most disabling chronic pain syndrome [1,2], characterised by abnormal and intense enhancement of pain perception with hyperalgesia, allodynia and receptive field expansion [3,4], usually associated with muscle and joint stiffness, insomnia, fatigue, mood disorders, cognitive dysfunction, anxiety, depression and intestinal irritability [5,6]. All these symptoms cause a significant impairment in the FM patients’ quality of life, with the inability to perform normal daily activities [5]. The American College of Rheumatology (ACR) has established criteria for FM diagnosis mostly based on two variables: (i) bilateral pain above and below the waist with centralised pain; and (ii) chronic generalised pain for at least three months. According to these criteria, pain is observed on palpation in at least 11 of 18 specific body sites [2,5]. FM especially affects women, with an estimated prevalence ranging from 0.2% to 6.6% [5] and with an age range onset between 30 and 35 years [2]. 

FM is currently classified under the group of central sensitivity disorders [3,7], which also include chronic fatigue syndrome, irritable bowel syndrome (IBS), temporomandibular joint dysfunction and tension headache. Notably, a high rate of comorbidity has been reported between FM and the other central sensitivity syndromes [8,9,10]. The International Association for the Study of Pain (IASP) has proposed a new classification for chronic pain [1], distinguishing between primary and secondary pain syndromes. FM has been included in musculoskeletal primary pain disorders, together with complex regional pain syndrome and nonspecific low back pain [1,11].

Stress and depression are considered as potent factors involved in the physiopathology of FM for their capability to dysregulate neuroendocrine, immune and pain mechanisms, resulting in various dysfunctions, such as motor impairment, cognition impairment, depression and long term pain [12,13,14]. The lifetime prevalence of depressive disorders in FM patients ranges between 40 and 80%, depending on the employed diagnostic criteria [12,14]. Moreover, FM may occur with other chronic inflammatory diseases such as rheumatoid arthritis, osteoarthritis and systemic lupus erythematosus [13,15].

Although the aetiology of FM is not completely understood, the involvement of several biological factors has been suggested, including abnormalities of the hypothalamic–pituitary–adrenal axis (HPA), dysfunction of autonomic nervous system, genetic factors, immunological alterations, oxidative stress, psychosocial variables and environmental stressors [4,5,13,16]. 

IBS is one of the most prevalent chronic gastro-intestinal diseases [17], mainly characterised by recurrent abdominal pain associated with alterations in either stool form or frequency, bloating, gas, diarrhoea or constipation [18]. Diagnosis is based on the current symptom-based criteria for IBS (Rome IV criteria), which must occur for at least 6 months [19]. This condition affects 4–10% of the global population and is associated with markedly reduced quality of life [20,21]. IBS is most common among women aged 20–40 years [22,23]. As for FM, IBS pathophysiology is still unclear and several factors has been implicated, such as genetic and environmental factors, mucosal immune dysfunction, intestinal dysmotility, increased intestinal permeability and visceral hypersensitivity [17,18]. Psychological stress and diet are also considered two important environmental factors closely linked to IBS. Previous acute enteric infections, observed in approximately 10% of IBS patients, represent another important factor contributing to predispose subjects to this syndrome [17,24]. All these factors might influence symptom severity. Of notice, IBS is also associated with common extra intestinal comorbidities including anxiety, depression, somatisation, insomnia, chronic fatigue and psychological disorders [25].

The coexistence between IBS and FM has been widely observed. Both FM and IBS are more prevalent in women [26] and are characterised by sympathetic dysfunction with central sensitisation [27,28]. A link between FM and IBS has been first postulated by Yunus et al. in 1981, who demonstrated that the prevalence of IBS in FM patients was approximately 50–70% [29]. Several further studies confirmed the high prevalence of IBS in FM and vice versa [30,31,32,33,34,35,36]. Accordingly, symptoms and signs of anxiety occur significantly more frequently in IBS patients than in controls and a sleep disturbance, typically associated with FM, has been described in up to 30% of IBS patients [30]. On the other hand, gastrointestinal symptoms in FM patients were reported to worsen during stress or disease exacerbations [37]. The coexistence of IBS and anxiety and depression has been observed in 30–35% of FM patients [34].

All these data clearly support the notion that FM is not only a musculoskeletal disorder but also shows signs of psychological as well as intestinal distress. However, the specific cause underlying this heterogeneity in symptomatology is not well defined. Alterations in gut microbiota and gut-brain axis, which connects the gut microbiota with the brain through the enteric nervous system, have been proposed as a possible FM pathogenetic mechanism [13,38]. In this review, we will discuss the alterations of microbiota and the gut-brain axis in FM and IBS patients, highlighting the similarities between these two syndromes and providing possible mechanisms involved in the physiopathology of FM. Based on therapeutic regimens used in IBS, we will also propose a possible therapeutic strategy to improve FM patients’ quality of life.

## 2. Human Microbiota and Gut-Brain Axis in Health and Disease

The human gut microbiota consists of a complex, dynamic and heterogeneous ecosystem inhabited by more than a trillion microorganisms including bacteria, archaea, fungi, viruses, protozoa and helminths interacting with each other and with the host [39,40,41]. With regard to the bacterial population, the human gut microbiota includes seven phyla: *Bacteroidetes*, *Firmicutes*, *Actinobacteria*, *Fusobacteria*, *Proteobacteria*, *Verrucomicrobia* and *Cyanobacteria*, with *Bacteroidetes* and *Firmicutes* representing more than 90% of the total bacteria [42]. The ratio between *Firmicutes* and *Bacteroidetes* is considered as an important parameter to take into account for the treatment of intestinal disorders [43]. The *Bacteroidetes* phylum includes *Bacteroides* and *Prevotella* genera, *Firmicutes* phylum includes *Clostridium*, *Eubacterium* and *Ruminococcus* genera [44]. Still, the relative richness of bacterial phyla may vary significantly among individuals [44]. The relationship between human host and gut microbiota is both commensal and mutualistic: while the host provides an ecological niche for all the components of the gut microbiota, some of them contribute to host development, fitness and metabolism. 

First of all, by living and replicating on intestinal surfaces, gut microbiota generates a stable system that prevents invasion of pathogenic microorganisms. In addition, gut microbes synthetise several classes of nutrients such as branched chain amino acids, amines, phenols, indoles, phenylacetic acid and vitamins [41,45,46,47]. Particularly, *Bacteroides* are involved in synthesis of biotin, riboflavin, pantothenate and ascorbate, while *Prevotella* are involved in thiamine and folate synthesis [44]. Gut microbiota contributes to the synthesis of bile acids, cholesterol as well as the absorption of calcium, magnesium and iron [46,48]. In addition, in stress conditions, it enhances the absorption of nutrients by increasing the length of intestinal villi and microvilli. 

Gut microbiota is considered the principal mediator of the metabolism of indigestible carbohydrates, such as cellulose, pectin and oligosaccharides, into short chain fatty acids (SCFAs) (acetate, propionate and butyrate), that are mainly produced by *Firmicutes, Bacteroidetes* and some anaerobic gut microorganisms [49]. They are rapidly absorbed by epithelial cells either by passive diffusion or active transport through G protein-coupled receptors such as GPR41, GPR43 and GPR109A [50]. SCFAs, particularly butyric acid and butyrate, are known to be fundamental for the maintenance of the intestinal barrier because of their capability to promote the expression of mucins, antimicrobial peptides and tight junction proteins [41,45,51,52]. 

SCFAs have also been demonstrated to possess anti-inflammatory effects. In particular, through the binding to GPR43, butyrate induces the production of anti-inflammatory cytokines such as TGFβ and IL-10 as well as the upregulation FoxP3, the master transcription factor of regulatory T cells (Tregs) [50]. Butyrate also inhibits histone deacetylase activity and downregulates the nuclear factor-κβ, one of the main mediators of the inflammatory response [50]. Furthermore, the combination between propionate and butyrate inhibits lipopolysaccharide (LPS)-induced inflammation by activating Tregs and reducing the production of inflammatory cytokines such as IL-6 and IL-12 [53].

Preclinical evidence also suggests that gut microbiota and its metabolites are involved in modulating behaviour and brain processes, including stress responsiveness, emotional behaviour and pain modulation [54]. Gut microbiota has been reported to be able to synthetise a range of neurotransmitters and neurotrophic factors, such as dopamine, noradrenaline, serotonin, gamma amino butyric acid (GABA), acetylcholine and histamine, that can affect the central nervous and peripheral enteric systems [40,55]. Signalling from enteric microbiota to the brain is mediated through epithelial-cell, receptor-mediated signalling and direct stimulation of the lamina propria cells [4]. On the other hand, the brain acts on enteric microbiota via changes in gastrointestinal motility, permeability and release of signalling molecules in gut lumen. This connection, known as the gut-brain axis, is extremely important to maintain the gastrointestinal homeostasis. 

The gut-brain axis is also involved in regulating neuronal, endocrine and immune pathways [38,40,56]. Therefore, a stable microbiota is critical for the maintenance of normal gut physiology and a proper transmission along the gut-brain axis. On the contrary, dysbiosis, i.e., the imbalance within gut microbial populations, negatively affects gut homeostasis and might cause an inappropriate activity of the gut-brain axis [43,57], as well as an impairment of central processing of sensory inputs [57,58]. Numerous risk factors have been proposed to be associated with the onset of gut dysbiosis: exposure to antibiotics and xenobiotics, such as heavy metals and pesticides, obesity, high-fat and high-sugar diets, host genetics, age and mode of birth [40,51]. 

Dysbiosis has been associated with the pathogenesis of many inflammatory diseases [17,25,51]. Moreover, alterations in the composition of the gut microbiota have been recently reported in FM [59,60]. Therefore, dysbiosis might represent an unfavourable condition contributing to FM development.

Together with dysbiosis, SIBO (small intestinal bacterial overgrowth) represents another type of qualitative and quantitative alteration of the gut microbiota that influences gut-brain axis communication [61]. In normal conditions, Gram-positive bacteria with 10^3^ organisms/mL mainly colonise the upper tract of the small intestine. On the contrary, during SIBO, the bacterial colonies increase to exceed 10^5^–10^6^ organisms/mL [62]. The human host controls the growth of enteric bacterial populations through several mechanisms. Indeed, gastric acids eradicate microorganisms, peristalsis sweeps the bacteria into the colon and their access is prevented thanks to the tight junctions between epithelial cells. Moreover, many antimicrobial products contribute to restraining bacterial overgrowth [63,64]. An impairment in one or more of those homeostatic defence mechanisms as well as certain anatomic abnormalities predisposes to SIBO development. Generally, patients with SIBO present nonspecific symptoms, such as bloating, abdominal distension, pain or discomfort, diarrhoea, fatigue, anxiety/depression and weakness [4]. Indeed, a similarity of symptoms between FM and SIBO has been observed, suggesting a possible role of SIBO in FM [65,66].

## 3. Microbiota Composition in FM Patients: Similarities and Differences with IBS

As previously mentioned, alterations in gut microbiota may affect the gut-brain axis [43,67]. Therefore, it is likely that dysbiosis might play a role in FM pathogenesis by altering perception and processing of painful stimuli [2,68]. Accordingly, analysis of gut microbiota in FM patients showed an altered composition [59,60]. Specifically, bacteria species belonging to the families of *Lachnospiraceae* and *Ruminococcaceae* as well as to *Eubacterium* and *Bifidobacterium* genera showed a lower abundance within the gut microbiota of FM patients, while *Rikenellaceae* family and many species belonging to the *Clostridia* class were overrepresented [59,60]. Many of the species whose abundance is altered in FM patients are involved in SCFAs metabolism. Indeed, *Lachnospiraceae* are involved in the synthesis of butyric acid, while *Eubacterium* species and *Faecalibacterium prausnitzii*, belonging to *Ruminoccaceae*, produce butyrate [53]. Thus, their depletion would suggest an impaired production of SCFAs, which in turn would negatively affect gut permeability. Since the major part of gut bacteria is Gram-negative-species shedding LPS, a leaky gut barrier may cause its systemic release. In the periphery, LPS can enhance pain perception either by directly interacting with peripheral neurons or by causing the broad activation of the immune system, which in turn secretes inflammatory mediators sensitising nociceptor neurons [69]. Moreover, SCFAs modulate the permeability of the blood–brain barrier by contributing to the correct organization of the tight junctions [70]. Therefore, in case of SCFA depletion, LPS could also reach the central nervous system (CNS) and act at the central level. Last but not least, SCFAs exert an anti-inflammatory activity by reducing leukocytes’ chemotaxis, adhesion and secretion of pro-inflammatory factors, thus counteracting the effects of LPS [71]. However, these beneficial effects are dose-dependent, since high concentrations of butyrate have been shown to promote apoptosis of intestinal cells, thus disrupting the intestinal barrier [72]. In FM patients, several SCFAs-producing bacteria of the *Clostridia* class have been found to be expanded [60]. In line with this observation, the concentration of butyric acid was increased in serum and urine of these subjects [60,68] supporting the hypothesis of a dysregulated SCFAs production in FM patients rather than a deficiency. 

On the other hand, bacteria from the *Bifidobacterium* genus participate in neurotransmitter metabolism by synthetizing GABA from glutamate [73]. GABA is the most important inhibitory neurotransmitter within CNS and acts by inducing neuron hyperpolarization and increasing excitability threshold, thus counteracting pain perception and transmission by nociceptive neurons. Conversely, glutamate acts in the opposite way and thus represents the major excitatory neurotransmitter involved in pain sensitisation [74]. As a consequence, a reduced presence of bacteria able to produce GABA, such as *Bifidobacterium*, would alter the GABA/glutamate balance in favour of the latter. Accordingly, peripheral levels of glutamate were found to be increased in FM patients [59]. Overall, this evidence suggests that the enhanced and diffused pain sensitivity observed in FM patients could involve a reduced capability of gut microbiota to produce GABA that, together with an increased permeability of the intestinal barrier, would in turn cause systemic accumulation of glutamate and widespread excitation of nociceptor neurons.

Bacterial species belonging to *Clostridia* class were also associated with disease severity symptoms, including widespread pain index, pain intensity, fatigue and sleep alterations [60]. Among *Clostridia* members, *Clostridium scindens* has been proposed to enhance pain sensitization because of its role in the production of bile acids. *C. scindens* is among the few species able to perform 7a-dehydroxylation needed for the conversion from primary to secondary bile acids [75], which has been proposed to participate in nociception [38]. Accordingly, secondary bile acids were found to be significantly altered in the serum from FM patients and to be associated with an increased presence of *C. scindens* and a generalised modification in the relative presence of bacterial species deputed to bile acid production in the gut. Particularly, a reduction in α-muricholic acid was reported, which is known to be degraded by *C. scindens.* Moreover, α-muricholic acid serum concentration negatively correlated with FM symptoms, indirectly supporting the possible pathogenetic role of *C. scindens* and bile acid alterations as a downstream mechanism in FM [76,77]. On the other side, bile acids are toxic for Gram-positive bacteria and induce the expansion of *Clostridia*, depleting beneficial species at the same time [78]. Thus, through a positive-feedback loop, bile acids might further enhance the gut dysbiosis observed in FM.

Interestingly, the alterations in gut microbiota composition observed in FM have also been reported in IBS (Table 1). *Ruminococcaceae* family, including *F. prausnitzii*, and *Bifidobacterium* genus have been shown to be reduced in IBS patients [52,79,80,81]. *F. prausnitzii* abundance negatively correlated with symptoms’ severity in IBS [82], in line with its role in protecting intestinal barrier through SCFAs production. Interestingly, in a non-inflammatory IBS-like rat model, disease symptoms and *F. prausnitzii* depletion were observed in animals experiencing stressful events in early life [83], strengthening the concept that neurotransmission can modulate gut microbiota composition through the gut-brain axis, which in turn affects the onset of painful stimuli. On the other hand, the bacteria from *Bifidobacterium* genus have been shown to exert several protective effects toward gut homeostasis, such as upregulation of tight junction proteins as well as downregulation of inflammatory mediators’ production from both intestinal and immune cells [84,85,86]. Therefore, the depletion of *Bifidobacterium* genus might contribute to the onset of intestinal symptoms in both IBS and FM. However, due its capacity to lower inflammation at systemic level [86] and to produce GABA [73], *Bifidobacterium* genus might also likely affect CNS. *Bifidobacterium* genus abundance has been demonstrated to be negatively associated with depression in IBS patients [87,88].

More conflicting evidence has been reported regarding *Lachnospiraceae*. An enrichment in this bacterial family was specifically observed in IBS patients with diarrhoea [89,90,91]. However, when gut microbiota in IBS patients was characterised regardless of intestinal symptomatology, a general depletion of *Lachnospiraceae* was reported [92,93,94]. Possibly, this discrepancy might be due to the enrichment/depletion of specific species within this family, which have not been characterised in detail in these studies. Of notice, low levels of *Lachnospiraceae* were reported in IBS patients showing anxiety and depression [93,95,96], which are common symptoms in FM [25], suggesting that *Lachnospiraceae* may be specifically involved in the onset of psychological distress observed in the two diseases. 

Although very few data are available about the increased abundance of *C. scindens* in IBS [97], the role of bile acids in the disease is otherwise well recognised. Increased levels of faecal bile acids have been reported in IBS patients, particularly those with diarrheic symptoms. Indeed, bile acids have been shown to be involved in several phenomena associated to diarrhoea, such as increased intestinal permeability, gut motility and abdominal pain [98]. Accordingly, *C. scindens* expansion has been specifically reported in diarrheic IBS patients [99].

**Table 1 biomedicines-11-01701-t001:** FM and IBS: similarities and differences in microbiota composition.

	Role	FM	IBS	References
*Bifidobacterium*	GABA synthesis	Reduction	Reduction	[53,60,80,81,82,83]
*Ruminococcoceae*	Production of butyrate	Reduction	Reduction	[53,60,61,80,81,82,83]
*Lachnospiraceae*	Synthesis of butyric acid	Reduction	Increase/reduction	[60,61,89,90,91,92,93,94,95,96]
*Eubacterium*	Production of butyrate	Reduction	Increase	[60,89,99]
*Rikenellaceae*	Digestion of crude fibre	Increase	Reduction	[60,90,91]
*C. scindens*	Production of bile acids	Increase	Increase	[77,78]

In contrast to FM (Table 1), the abundance of *Eubacterium* genus in IBS patients has been recently found to be increased in IBS and to correlate with severity symptoms, similarly to *Lachnospiraceae* [89,99]. On the other hand, *Rikenellaceae*, which are expanded in FM, are usually depleted in IBS [90,91], although some authors correlated their abundance with psychological symptoms [95].

Quantitative alterations within gut microbiota have also been reported in FM. Indeed, the major part of FM patients has been found to be tested positive for SIBO, as assessed by a lactulose hydrogen breath test [65,66]. SIBO incidence was higher in FM compared to IBS patients and correlated with pain severity [66], while the usage of antibiotics relieved intestinal symptoms in both FM and IBS [65,100]. It has been proposed that the expanded overall bacterial population could cause the massive translocation of bacterial endotoxins through a damaged intestinal barrier, resulting in the increased inflammation and hyperalgesia shared by FM and IBS [39]. However, FM patients tended to produce more hydrogen than IBS ones [66], suggesting that, together with general bacterial increase, the expansion of certain species involved in pain sensitisation might specifically occur in FM. 

Overall, this evidence indicates that gut dysbiosis might be a common leading cause for the onset of both FM and IBS. Dysbiosis together with SIBO is involved in the pathogenesis of FM and IBS and similarities in gut microbiota alterations could explain the two diseases’ overlapping symptoms. 

## 4. Discussion

FM is a disabling chronic pain syndrome, characterised by chronic pain frequently associated with chronic fatigue, sleep disturbances, cognitive dysfunctions as well as signs of psychological distress such as depression, anxiety and stress-related symptoms [1,2,5,101]. Numerous rehabilitation programs are currently being used to restore the compromised functions and improve the quality of life with poor results.

IBS is mainly characterised by abnormal pain and altered and irritable bowel as well as gastroesophageal reflux, oesophageal hypersensitivity and functional dyspepsia [18]. IBS is also commonly associated with extra intestinal comorbidities including anxiety, depression, somatisation, insomnia and chronic fatigue. All these clinical conditions are mostly associated with FM [25]. The aetiology of both FM and IBS is currently not completely understood, but a lot of hypotheses are developing. The literature supports the coexistence between IBS and FM, which suggests the existence of a common pathogenic mechanisms for both conditions [26,29,30,31,32,33,34,35,36,37].

Gut microbiota represents a heterogeneous ecosystem composed of billions of microorganisms that play a fundamental role for host health [39,40,41]. Some of the beneficial effects exerted by intestinal microbiota occurs through the gut-brain axis, which implies that any alteration or distress in gut microbiota homeostasis can affect gut-brain axis regulation mediated through immunological, hormonal and neural pathways [56,67,102,103]. Alterations in this axis have been associated with gastrointestinal syndromes [17,25,51]. Recently, the possibility that gut microbiota could contribute to regulation of chronic pain has attracted more attention. Literature supports the fact that the gut microbiota is involved in the central sensitisation of chronic pain as well as inflammatory diseases such as endometriosis by regulating microglia, astrocytes and immune cells [101,104]. Therefore, knowing the role of the human gut microbiota in the pathogenesis of pain could open a possibility to use it as a possible target for analgesic therapies. 

The causes underlying the heterogeneity in FM symptoms are not well defined. Among the possible pathogenetic mechanisms, alterations in gut microbiota and the gut-brain axis have been proposed. SIBO as well as alterations in gut microbiota balance have indeed been reported [59,60,65,66,68]. Experimental findings showed that gut dysbiosis in FM and IBS patients share several features in terms of abundance/depletion of bacterial families and genera [52,59,60,79,80,81,92,94,95,96]. This evidence, together with the symptoms overlap, are suggestive of a common origin for FM and IBS, which might actually represent different manifestations of the same pathological entity. This hypothesis could also justify the apparent discrepancies reported regarding gut microbiota in the two groups of patients. It is possible that, in the context of a common frame of central distress, alterations in specific species might modulate the clinical manifestations of the disease, shifting the balance toward pain sensitization or intestinal symptoms.

Nowadays, the therapeutic approaches for the management of chronic pain are widely investigated [105]. The most prominent approach could be represented by probiotic intervention, defined as live microorganisms whose administration confers a health benefit to the host, such as improved digestion, boosted immunity and decreased cholesterol levels, all associated with lowered risk of certain diseases [104,106,107]. The benefits of probiotics in IBS patients have been studied in depth. *Lactobacillus, Enterococcus* and *Bifidobacterium bifidum* were shown to improve symptom severity in IBS [108,109]. Butyrate producers such as *Faecalibacterium* sp. have an anti-inflammatory effect on the gastrointestinal tract. *F. prausnitzii*, as a source of serine protease, is able to decrease the excitability of dorsal root ganglia neurons with an anti-nociceptive activity [110,111]. *Bifidobacterium longum* was associated with a significant reduction in depression and an increased quality of life in IBS patients, but no changes in IBS symptoms severity or faecal microbiota profile were reported, suggesting that *B. longum* might act at the CNS level [112]. The beneficial effects of *Clostridium butyricum* and *Roseburia hominis* against visceral hypersensitivity in IBS were observed in preclinical animal studies [113,114]. *Lactobacillus* and *Bifidobacterium* bacteria are capable to prevent the chronic stress-mediated brain function abnormalities by modulating the HPA axis response [115]. Indeed, a probiotic mixture of *B. infantis* and *B. longum* in IBS children has been demonstrated to improve abdominal pain and quality of life [116].

Probiotics have been shown to increase the production of SCFAs, which have been associated with peripheral nerve sensitisation and abdominal pain relief [117]. This evidence indicated that SCFAs may be important gut microbiota mediators in the regulation of pain through receptor-mediated mechanisms in IBS patients. Probiotics can also affect the inflammatory response by modulating pro-inflammatory and anti-inflammatory cytokines and possible analgesic effects of probiotics on inflammatory pain are expected [107]. Another mode of action of probiotics is to regulate pain through gene expression of pain-related receptors on epithelial cells [107]. For example, *L. acidophilus* is known to increase the expression of cannabinoid receptor 2 and colonic μ-opioid receptor together to reduce pain sensation [118]. All these data indicate that the use of probiotics is indicated as a potential treatment to cure IBS symptoms and could be also considered as a potential strategy to treat chronic pain [104].

The use of probiotics in the treatment of FM was also studied. *L. casei* and *B. infantis* showed the capability to improve cognition, particularly impulsive choice and decision-making in FM patients. However, no other beneficial effects were observed in self-reported pain, quality of life, depression or anxiety [119]. The role of probiotics in the improvement of FM cognitive processes was assessed by using a combination of *L. rhamnosus, L. paracasei, L. acidophilus* and *B. bifidus*, reporting a beneficial effect on cognitive and emotional symptoms [120]. However, such a benefit has not been confirmed, probably because of the short length of the treatment [120]. On the contrary, the efficacy of probiotics treatment has been observed in Alzheimer disease, demonstrating an improvement in learning and memory [121]. 

Since SIBO has been found to be widely associated with FM, the usage of antibiotics as treatment has also been proposed. Accordingly, antibiotic therapy in FM patients has been demonstrated to be useful to counteract intestinal distress [65,100]. However, such a finding has not been further investigated in clinical trials. 

Overall, current data are insufficient to evaluate the utility of probiotics and/or antibiotics in FM therapy, and more research will be required to confirm the effectiveness of these interventions.

Faecal bacteriotherapy or stool/faecal transplantation (FMT) is the infusion or engraftment of liquid filtrate faeces from a healthy donor into the gut of a recipient. FMT is used to treat *Clostridium difficile* infection, IBD, obesity and insulin resistance [107]. A study of FMT in IBS patients showed improvement in abdominal pain associated with the relative abundance of *Akkermansia muciniphila* [122]. FMT studies in rats showed that visceral hypersensitivity was induced by the transplantation of the faecal microbiota from constipation-predominant IBS patients [123].

FMT represents a possible therapeutic strategy to improve FM patients’ lives. In fact, FM patients were reported to be in full recovery after FMT [124]. In this study, an increase in faecal *Bifidobacterium* proportion from 0% to 5.23% and a reduction in *Streptococcus* from 26.39% to 0.15% was observed [124]. The molecular mechanisms underlying gut microbiota capability to modulate pain should be investigated in the future and could be useful for the discovery of novel drugs for pain relief. It is well elucidated that gut microbiota may also play a role in depression and anxiety, which co-exist with pain syndromes. Thus, the modulation of gut microbiota for the control of pain should be addressed. FMT should be considered as a valuable therapeutic option for the cure of pain. Indeed, FMT may become a promising approach to treat FM patients, thus reestablishing the microbiota gut-brain axis and limiting the symptoms associated with IBS as well as pain relief. 

Collectively, current findings show that gut microbiota composition is quantitatively and qualitatively altered in FM and IBS patients, which suggests an active role of intestinal bacterial species in modulating several common aspects of the two pathologies such as pain sensitisation, intestinal symptoms and physiological manifestations. However, whether such a dysbiosis causes FM/IBS onset or it is rather a consequence of alterations at central level affecting the gut-brain axis is not known. Studies investigating gut microbiota composition have been conducted in subjects with confirmed pathology, which allowed the assessment of the association between FM/IBS and the relative abundance of specific microbial species, but not to define the causal link. However, evidence provided by FMT studies favour the first hypothesis. Intestinal microbes from patients induced IBS symptoms when transplanted in germ-free mice [91], while FMT from healthy donors to IBS patients was able to relieve psychological symptoms [125]. Similarly, neurobehavioral changes were observed in rats when FMT was performed by using patients with depression [126]. Overall, this evidence indicates that dysbiosis might be an up-stream mechanism determining the onset of symptoms, particularly neurological ones, observed in FM and IBS. Further studies in animal models as well as a more in-depth knowledge of dysbiosis at species level will provide more insights about the specific pathological mechanisms induced by bacteria expansion/contraction and the cause–effect relationships linking gut microbiota alterations to the heterogeneous symptoms of both FM and IBS. This knowledge will be in turn useful to design tailored therapeutic strategies to restore specific bacterial populations in order to relieve the most debilitating symptoms in FM patients. Still, whether the dysbiosis is a leading cause or a consequence of other pathological mechanisms in both FM and IBS is still unknown.

## 5. Conclusions

IBS and FM are defined as central sensitivity disorders in which patients perceive pain by hyperalgesia and receptive field expansion. The pathophysiological mechanisms underlying FM and IBS remain, to date, still underexplored, although some hypotheses have been proposed. Among them, the role of gut-brain axis as well as gut microbiota alterations (dysbiosis, SIBO) have gained much attention. Indeed, increasing reports are describing the expansion/contraction of specific bacterial species in gut microbiota of both FM and IBS patients as well as the association between these alterations and symptoms severity. Notably, several of these changes have been reported in both the diseases, providing a biological mechanism underlying the co-occurrence and symptoms overlapping of the two pathological entities and supporting the hypothesis that they may represent different manifestations of the same disease. Accordingly, the use of probiotics as well as FTM could be prominent strategies to improve symptoms severity of FM and IBS, with a reduction of depression and anxiety, as well as pain relief. However, more studies are needed to effectively demonstrate the efficacy of these interventions, particularly in FM patients.

To date, there is no clear evidence about the causative link between dysbiosis and both FM and IBS, although FMT in animal models strongly suggest that alterations in gut microbiota occur before clinical onset and are sufficient to cause at least some of the typical symptoms common to FM and IBS. Further studies, particularly analysis of gut microbiota composition in healthy subjects and follow-up studies, are necessary to formally prove the causative role of gut microbiota, and in turn, of the gut-brain axis in FM and IBS physiopathology. Such a knowledge will be useful to improve current approaches, design new therapeutic regimens and, possibly, develop preventive strategies for high-risk subjects. 

## Data Availability

Not applicable.

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
