# Peer review of "Fibromyalgia and Irritable Bowel Syndrome Interaction: A Possible Role for Gut Microbiota and Gut-Brain Axis"

_biomedicines, 2023, doi:10.3390/biomedicines11061701_

Round 1
Reviewer 1 Report
Biomedicines #2346765
Fibromyalgia and Irritable Bowel Syndrome Interaction: a possible role
for gut microbiota and gut-brain axis
We have completed review the manuscript" Biomedicines #2346765". This review described that possible role for alternation of gut-microbiota contributes to Fibromyalgia and Irritable bowel syndrome.
Although description presented here are an interesting and well written, this reviewer recommend that authors should add the conclusion paragraph of the end of manuscript for readers.
Author Response
We thank the Reviewer for the useful comment. According to his/her suggestion, we added a conclusion paragraph at the end of the manuscript (Pages 9-10 lines 446-471)
Reviewer 2 Report
I have read the submitted manuscript and provide the following comments and queries for the authors to address.
Check that the nomenclature and presentation of bacteria is consistent throughout the manuscript. Some journals require that all bacterial designations should be italicized. Check with the journal's style if this is so for the current journal.
In section 3 there was suggested that adverse effects such as loss of epithelial barrier integrity, pro-inflammatory activities and metabolic endotoxemia were present in both FM and IBS conditions...was intestinal dysbiosis as observed by the authors the primary trigger of both FM and IBS ? Clarify.
In table 1 the authors present butyric acid (which is the conjugate base of butyrate) and butyrate clarify the meaning of these 2 forms.
It is also possible that bile acids affect the gut-brain axis and dysregulation of bile salts can further exacerbate intestinal dysbiosis. This could be further discussed re FM and IBS.
As I understand it, this review is largely a narrative review and as such these reviews have several limitations that should be discussed as well as provide any relevant future developments.
Author Response
We thank the Reviewer for highlighting this point. We amended bacteria nomenclature according to the journal's style. The Reviewer raises a very important point. Actually, association between dysbiosis and both FM and IBS has been observed but the causative relationship, i.e. whether the dysbiosis is the cause of the consequence of FM and IBS has not been formally proven. However, some experimental data point to the first hypothesis. We provided some hints about this point in the Discussion, but, according to the Reviewer's suggestion, we further expanded this section (Page 9 Lines 421-443) with more experimental data and a more in-depth discussion of the limits of the current studies in human subjects that do not allow a clear definition of the causative link between dysbiosis and FM/IBS. We thank the Reviewer for the comment. Although the two terms are often used as synonyms, reviewing the literature it is clear that specific bacterial species preferentially produce one of the two form of the molecule. For example, in the human gut, several members of the Lachnospiraceae family are involved in butyric acid synthesis. On the contrary, mostly several Eubacterium species together with Faecalibacterium prausnitzii, belonging to Ruminoccaceae, produce butyrate (ref. 49,60,61). In light of this, we preferred to differentiate between the two molecules in the text. We thank the Reviewer for the useful suggestion. Accordingly, we discussed this point in Paragraph 3 (Page 5 Lines 236-248 and Page 6 Lines 280-288 ) We thank the Reviewer for the comment. We tried to improve our Discussion, particularly the points related to therapies and causative link between dysbiosis and FM/IBS, and added a conclusion paragraph to summarise the relevant findings obtained by literature survey.
Reviewer 3 Report
Currently, the participation of gut microbiota in the pathogenesis of fibromyalgia and IBS is of great interest. Dysbiosis results in gut dysmotility, vesceral hypersensitivity, and peripheral and central neurotransmission disorders. For this reason, antibacterial agents or probiotics, and faecal transplantation have been used, but unambigueus results have not been obtained, and further research, including nutritional interventions, is needed. Probably, not only alteration in composition of intestinal bacteria, but also production modulation of many biological active compounds in the GIT, including neurotransmitters and neuromodulators, may be the key to reducing psychosomatic ailments. For example, some clinicians in the treatment of FM recommended reducing the intake of tryptophan and HT-3 inhibitors, while other use drugs that inhibit serotonin reuptake in tissues.
This work does not cover all aspects of FM and IBS coexistence, but raises very important and interesting issue. The articlr is well prepared in terms of content and editorial and can be published in the presented version.
Author Response
We warmly thank the Reviewer for the positive evaluation of our work.